# The Impact of Upper Limb Apraxia on General and Domain-Specific Self-Efficacy in Post-Stroke Patients

**DOI:** 10.3390/healthcare11162252

**Published:** 2023-08-10

**Authors:** Laura Sánchez-Bermejo, Pedro Jesús Milla-Ortega, José Manuel Pérez-Mármol

**Affiliations:** 1Department of Physiotherapy, Faculty of Health Sciences, University of Granada, 18071 Granada, Spain; lasai@correo.ugr.es; 2Instituto de Investigación Biosanitaria ibs.GRANADA, 18012 Granada, Spain; pedroj.milla.sspa@juntadeandalucia.es; 3Emergencies Primary Care Service, Granada Health District, 18012 Granada, Spain

**Keywords:** self-efficacy, self-efficacy for managing symptoms, apraxia, upper limb apraxia, stroke

## Abstract

Background: Upper limb apraxia (ULA) is a neurological syndrome characterized by the inability to perform purposeful movements. ULA could impact individuals’ perceptions, including perceived self-efficacy. The aim of this study is to investigate whether ULA is related to general self-efficacy and self-efficacy for managing symptoms in post-stroke patients. Methods: A cross-sectional study was conducted involving 82 post-stroke patients. Regression analyses were implemented using a stepwise model including seven dimensions of ULA: imitation (non-symbolic, intransitive, and transitive), pantomime (non-symbolic, intransitive, and transitive), and dimension of apraxic performance in activities of daily living. These dimensions were independent variables, while general self-efficacy and symptom management self-efficacy dimensions were dependent variables. Results: The findings revealed that intransitive imitation accounted for 14% of the variance in general self-efficacy and 10% of self-efficacy for managing emotional symptoms. Transitive imitation explained 10% of the variance in self-efficacy for managing global symptoms and 5% for social–home integration symptoms. The combination of intransitive imitation, non-symbolic pantomime, and alterations in activities of daily living performance associated with ULA explained 24% of the variance in cognitive self-efficacy. Conclusions: Hence, ULA dimensions seem to be related to the levels of general perceived self-efficacy and self-efficacy for managing symptoms among post-stroke patients.

## 1. Introduction

Upper limb apraxia (ULA) is a neurological syndrome that has a substantial impact on the functional interaction between individuals and their environment. The main clinical symptom of ULA is the incapacity to perform deliberated movements upon a request from a person or the context, especially in hands. ULA is usually considered a heterogeneous syndrome historically defined by the exclusion of several health parameters, ordered by frequency in the literature: (i) motor and sensory impairments, (ii) comprehension deficits, (iii) weakness, (iv) coordination impairments, (v) intellectual deterioration, (vi) uncooperativeness, (vii) lack in motivation, (viii) cognitive deficits (memory and attention), (ix) movement alteration (tremors, chorea, athetosis, myoclonus, and dystonia), (x) disorder of tone or posture, and (xi) object recognition difficulties [1,2,3]. ULA is caused by different types of brain damage, but vascular aetiology seems to have the most prevalence [4]. Apraxia is closely linked to damage in key regions such as the parietal lobe, premotor cortex, and underlying white matter [2,5,6]. Approximately 25% of stroke survivors also have ULA [7], and 50% of the individuals with left stroke exhibit apraxia that persists following illness onset [7,8].

The primary clinical manifestation of ULA is the inability to execute deliberated, voluntary, and purposeful movements in response to a request from an individual or the environment. These movements may be classified based on the nature of the action performed. Transitive movements involve manipulating objects, while intransitive movements are performed without using objects. Symbolic movements are those that have social or cultural significance, whereas non-symbolic movements do not have any such significance. The main sensory pathways through which information required to configure a movement is typically received include the auditory and visual modalities [9]. On the other hand, the completion of a movement involves the integration of all relevant internal and external information, including the physical characteristics of objects in the real world. This information is commonly stored within memory, whether consciously or unconsciously, and is referred to as action semantic knowledge. This knowledge in ULA is related to the ability to use stored knowledge of tool manipulation and the properties of objects to perform purposeful movements. Additionally, it is important to consider the quality of movements exhibited during the performance of daily activities in a natural or real-life context. ULA can have an impact on daily living, causing difficulty in performing skilled movements necessary for tasks, such as buttoning clothes or brushing teeth. These difficulties could decrease functioning independence [10,11].

The classification of apraxia has historically been divided into two main groups, ideomotor and ideational apraxia; however, contemporary research understands this syndrome in a broader and more complex manner. A deeper comprehension of ULA requires an analysis that integrates the assessment of apraxic errors with a diverse range of gestures that represent a wide spectrum of upper limb movements [12]. These gestures encompass a range of features, such as the distinction between symbolic and non-symbolic gestures, transitive and intransitive movements, as well as imitation and pantomime. Pantomime is regarded as the capacity to execute a gesture in a simulated context, that is, a context different from the one in which the gesture is normally performed, without having access to sensory information specifically related to that gesture. Moreover, they also differ in terms of the location of the movement (proximal or distal) and the complexity of the movement (simple or repetitive) [13]. The main errors that define ULA include errors in movement content (e.g., making the ‘stop’ gesture when asked for the ‘military salute’), spatial (e.g., bringing a spoon to the nose instead of the mouth), timing (e.g., making the gesture slower than expected), or unrecognizable actions [12]. Generally, the prevalent errors arising from ULA are those associated with the content of actions in both transitive and intransitive gestures [14].

General self-efficacy is commonly defined as the comprehensive belief in one’s capability to effectively perform actions aimed at managing personal and everyday life goals. Bandura characterized self-efficacy by four primary sources: direct mastery experiences, vicarious experiences, verbal persuasion, and physiological state [15]. These primary sources can have an impact on emotions, thoughts, and behaviours. This impact is directly related to the health concept and performance in health. Self-efficacy modulates the effort in achieving health goals, treatment adherence, and resilience in difficult contexts [16]. In people who have suffered a stroke, self-efficacy can significantly impact the recuperation process. Heightened levels of self-efficacy are correlated with increased initiation and engagement in activity performance, leading to the achievement of clinical goals [17]. This association may be attributed to the relationship between self-efficacy with improvements in mobility, balance and fall risk, community reintegration [18], activities of daily living, perceived health status, depression [16,19], frailty progression [19], and quality of life [20]. Individuals diagnosed with stroke may encounter inconsistencies or incongruence in their sense of self, both pre-and post-lesion. This incongruence can impact their levels of self-efficacy, which may not be included in their rehabilitation plans [21].

Domain-specific self-efficacy for managing symptoms is the ability to perform actions to achieve specific goals related to different types of symptoms. These goals could be recognizing and adapting to the symptoms, evaluating risk situations, and preventing health complications [22]. Domain-specific self-efficacy is helpful to the active role of people who have suffered a stroke [23]. The World Health Organization supports the patient’s autonomy in plans for symptom management. The treatment of domain-specific self-efficacy could facilitate autonomy and decision making in illness control [22]. Studies in the last few years have found that self-efficacy influences health status, as well as physical and psychological symptoms in chronic illness [24,25,26,27]. In people with stroke, higher levels of domain-specific self-efficacy could predict better results in health [27]. Self-efficacy levels are tightly related to behaviour, affecting self-management [28].

The presence of ULA often leads to difficulties in performing activities of daily living, which can result in perceptions of dependence and limitation [29,30]. Furthermore, ULA can also affect non-verbal communication, leading to deficits in social skills and a reduced ability to interact with the environment [31]. The conscious and unconscious identification of these handicaps when engaging in environmental interaction can have an impact on an individual’s self-concept. When self-concept is affected, it is common for individuals to perceive frustration, inefficiency, and decreased self-esteem [32,33]. The combination of this perception as well as reduced environmental interaction may decrease the ability of the individual to confront daily stressors [34]. The perception of how an individual faces daily challenges, demands, and stressors is determined by levels of general self-efficacy and self-efficacy for managing symptoms in people with health conditions [25]. Since post-stroke patients manifest ULA, it is plausible to hypothesize that higher levels of ULA are associated with lower levels of self-efficacy. Therefore, the aim of this study is to investigate whether ULA is related to the levels of general self-efficacy and self-efficacy for managing symptoms in post-stroke patients.

## 2. Materials and Methods

### 2.1. Study Design

A cross-sectional, observational, and descriptive study was conducted. The participants were post-stroke patients recruited from the Andalusia public health system.

### 2.2. Participants Selection and Sample Size Estimation

The inclusion criteria for the participants were (i) having mild to moderate sequelae assessed using the National Institutes of Health Stroke Scale (NIHSS) [35], (ii) being older than 18 years old, and (iii) having native or fluent proficiency in Spanish to understand the assessment instructions. The exclusion criteria were (i) individuals with a history of non-vascular brain damage, (ii) neurodegenerative disease, (iii) moderate–severe cognitive decline [36], (iv) severe intellectual disability, (v) diagnosis of a severe mental disorder, (vi) a musculoskeletal disorder, (vii) peripheral nervous system injuries, (viii) uncorrected sensory impairments, and (ix) impairments in communication. 

The sample size for this study was determined using G*power. A priori computed required sample size estimation was conducted using a *t* test family and a linear multiple regression fixed model single regression coefficient including seven predictors, one-tailed. Considering a moderate effect size of f^2^ = 0.15 [37,38], a desired statistical power of 95%, and a significance level of 0.05, the calculated sample size was 74 participants. To account for potential losses during the study, a 10% increase in the sample size was implemented, resulting in a total sample of 82 post-stroke patients.

A total of 189 post-stroke patients met the criteria for suitability and agreed to participate in the study. From this group, 82 participants were selected, with 41 individuals having ULA and 41 without ULA. ULA was evaluated using the Apraxia Screen of TULIA (AST) test, where a score of <9 points indicates the presence of apraxia [39]. A matching process was implemented to ensure that the 82 participants of the study represented a balanced distribution of people with and without ULA. The post-stroke patients labelled as ‘non-ULA’ usually exhibited apraxic errors, even if they did not exceed the established test cut-off point. The errors committed by the whole sample of post-stroke patients included in the study are representative of the variability in the praxis function continuum registered in the ULA tests [40]. The flowchart of the study participants’ selection is depicted in Figure 1.

### 2.3. Study Procedures

Patients were gathered from randomly selected primary care centres during the period of April 2022 and June 2023. Prior to their appointments, they were provided with detailed information about the study. Individuals who expressed interest in participating were given an informed consent form and a participant information sheet. Afterwards, the evaluation of ULA and self-efficacy was implemented during a one-and-a-half-hour session. The evaluation system was applied by a researcher who has specific training in evaluations of post-stroke patients and upper limb apraxia and more than ten years of clinical expertise in upper limb apraxia. The data collection and evaluation process was implemented in the following order: first, sociodemographic and clinical information was collected; second, the general self-efficacy and the self-efficacy in symptom management scales were administered; and finally, the TULIA test and the ADL observation scale were implemented. 

The Ethics Committee for Biomedical Research CEI-Granada in the Province of Granada (Andalusia, Spain) approved this study with reference: 1503-N-21. The research study adhered to the ethical guidelines outlined in the Declaration of Helsinki throughout the entire research process, including data collection.

### 2.4. Evaluation System

#### 2.4.1. Socio-Demographic, Lifestyle Behaviour, and Clinical Data

Researchers gathered information on socio-demographic data (age, gender, marital status, occupation, and educational level), self-reported lifestyle behaviours (smoking, alcohol and caffeine consumption, hours of sleep, and physical activity), clinical information (dominant hand, body mass index, occupational therapy rehabilitation, and hemiplegia), and stroke characteristics (stroke type, hemispheric stroke, and time since stroke).

#### 2.4.2. Self-Efficacy Dimensions

##### General Self-Efficacy

The general self-efficacy scale was utilized to assess general self-efficacy in the study. This scale consists of 10 items that measure an individual’s self-perceived ability to cope with stressful situations. Each item is rated on a 10-point Likert scale, ranging from 0 (indicating ‘completely disagree’) to 10 (indicating ‘completely agree’). Higher scores on the scale indicate higher levels of self-efficacy. The psychometric properties of the scale in the Spanish population demonstrate significant internal consistency, with alpha coefficients ranging from 0.79 to 0.93, indicating a reliable measure of self-efficacy. Additionally, the scale exhibits strong correlation with other measures of self-efficacy, with a correlation coefficient of 0.88 [41]. The study participants exhibited a strong level of internal consistency, with a Cronbach’s alpha coefficient of 0.86, indicating high reliability. 

##### Domain-Specific Self-Efficacy for Managing Symptoms

Self-efficacy for managing symptoms was evaluated using the self-efficacy in symptom management scale after the traumatic brain injury. This is a specific questionnaire that evaluates self-efficacy after brain damage. It contains 13 items divided into three dimensions: perception of self-efficacy for managing social and community situations (social–home integration self-efficacy), management of physical or cognitive symptoms (cognitive self-efficacy), and management of emotional symptoms (emotional self-efficacy). The internal consistency of the scale for the Spanish population showed notable internal consistency. Social–home integration, cognitive and emotional dimensions, and the total score of the scale showed a Cronbach’s alpha coefficient of 0.66, 0.93, 0.89, and 0.89, respectively [42].

#### 2.4.3. Construct of Upper Limb Apraxia

##### Domains of Imitation and Pantomime

The Comprehensive Assessment of Gesture Production (TULIA) test was used to evaluate the imitation and pantomime dimensions. The TULIA comprises 48 items that are categorized into six differentiated dimensions (sub-constructs of apraxia): non-symbolic imitation (meaningless); intransitive imitation (communicative); transitive imitation (tool related); non-symbolic pantomime; intransitive pantomime; and transitive pantomime. These items assess various kinematic features, including proximal and distal movements, as well as simple and repetitive actions. Each item is scored on a scale ranging from 0 to 5, with a score of 5 indicating normal movement. The total score ranges from 0 to 240, with lower scores indicating a greater presence of ULA. For stroke patients, the authors established a cut-off score of 194 points to identify the presence of ULA. The TULIA test has shown adequate psychometric properties, including internal consistency (with Cronbach alpha values ranging from 0.67 to 0.92), and high intraclass correlation (0.96 for imitation and 0.99 for pantomime) [13].

##### Domains of Apraxic Performance in Activities of Daily Living (ADL)

The ADL observation scale is a validated unidimensional tool specifically designed to assess the ADL performance associated with ULA. The evaluation is based on the observation of the performance of four daily activities, three of which are previously established, and the other is chosen by the evaluator. The scoring method consists of four aspects: independence, initiation, execution, and control. Higher scores indicate greater dependence on these activities, reflecting the impact of ULA. The individual scores for each activity can be summed to obtain a total score, providing a comprehensive measure of ULA-related disability. The ADL observation scale demonstrates strong internal consistency, with a Loevinger’s H-coefficient of 0.58 and a rho-value of 0.94, indicating its reliability in measuring ULA. Moreover, the tool exhibits high inter-observer reliability, as evidenced by strong agreement between observers, with an intra-class correlation coefficient of 0.98, particularly for the total score of the assessment [10].

### 2.5. Statistical Analyses

The collected data were analysed using SPSS version 26.0. To gain insights into the dataset, descriptive statistics were employed to summarize the key variables, including means, standard deviations, frequencies, percentages, ranges, minimums and maximums, and percentiles), To perform a between-group comparison, the independent *t*-test and Chi-Square test were run for continuous and categorical variables, respectively. To identify if ULA dimensions are associated with self-efficacy dimensions, five separate multiple linear regression analyses were performed with the total sample of post-stroke patients (n = 82). The stepwise regression approach was used to select the significant predictors for each self-efficacy dimension, considering the seven ULA dimensions as potential predictors. Specifically, dependent variables were general self-efficacy and self-efficacy for managing symptoms dimensions: social–home self-efficacy, cognitive self-efficacy, emotional self-efficacy, and the total score of self-efficacy for managing symptoms. The independent variables comprised the seven dimensions of ULA, that is, the six dimensions of the TULIA test (non-symbolic imitation, intransitive imitation, transitive imitation, non-symbolic pantomime, intransitive pantomime, and transitive pantomime), and the total score of the ADL observation scale. Before conducting the regression analyses, the normality of the variable distributions and the presence of multicollinearity were assessed. The variance inflation factor (VIF) was utilized as an indicator of collinearity, and normality was evaluated using appropriate statistical tests. Statistical significance was determined using a *p*-value threshold of less than 0.05.

## 3. Results

### 3.1. Sample Description

Table 1 shows the descriptive statistics of the 82 participants selected in the study, with and without ULA. No significant differences were found between the groups in any sociodemographic, clinical, or lifestyle behaviour data. The findings indicate that a significant proportion of participants were men, accounting for 73% of the total sample, with a mean age of 62 years. The most frequent dominant hand was the right, and only 5% of the sample had hemiplegia. Ischemic stroke accounted for 87% of the cases and occurred, on average, 31 months prior to the study.

The descriptive statistics for the seven dimensions of ULA are shown in Table 2. The mean scores ranged from 29.55 to 36.68 points across different dimensions. The dimension of ‘transitive imitation’ shows the lowest mean score. The percentiles exhibit information about the distribution of scores within each dimension, reflecting variations in the performance test within the continuum of praxic function.

### 3.2. The Influence of ULA Dimensions on Self-Efficacy Dimensions

Regarding self-efficacy dimensions, results after linear regression analyses have shown that intransitive imitation dimension was significantly associated with general and emotional self-efficacy, explaining 14% (*p* < 0.001) and 10% (*p* = 0.003) of its variance, respectively. Transitive imitation explains 10% (*p* < 0.001) and 5.2% (*p* = 0.035) of the variance in self-efficacy for managing symptoms and social–home integration. Intransitive imitation, the total score of the ADL observation scale, and non-symbolic pantomime explained 24% (*p* = 0.001; *p* = 0.009; *p* = 0.028, respectively) of the cognitive self-efficacy variance. These results are shown in Table 3. 

## 4. Discussion

This study examines if upper limb apraxia dimensions are related to the levels of general perceived self-efficacy and self-efficacy for managing symptoms among post-stroke patients. The results indicated that the intransitive imitation was associated with general self-efficacy and self-efficacy for managing emotional symptoms. The transitive imitation was related to self-efficacy for managing global, social, and home integration symptoms. Finally, the intransitive imitation, non-symbolic pantomime, and ULA-associated alterations in ADL performance explained the dimension of cognitive self-efficacy.

The imitation of intransitive gestures, that is, the capacity for gesture replication without using an object, was shown to be related to several aspects of perceived self-efficacy in the sample of post-stroke patients. This finding may be explained because the loss of control and ability to manage daily activities can present challenges related to self-efficacy [43]. Self-efficacy, in turn, plays a mediating role in influencing occupational performance and occupational performance satisfaction [44]. Apraxic errors in gesture execution independent of objects may contribute to these challenges. This type of gesture is involved in communication tasks, interaction with the environment, and daily routines. Intransitive gestures possess a strong social component as they are commonly used while communicating and are highly influenced by contextual factors [45]. These errors in imitating intransitive gestures may be associated with the individuals’ perception of their own ability to respond to external demands [46].

The imitation of transitive gestures, that is, the ability for gesture imitation using an object, was associated with self-efficacy for managing global, social, and home integration symptoms, probably due to several reasons. Imitation allows for the replication and learning of movements that can be useful in daily life [47]. When imitation involves objects, individuals focus their attention on how the objects are used rather than the specific details of the movements themselves [48]. Transitive imitation leads individuals to execute the most effective movements to achieve the final goal in a real context where the use of instruments, objects, or materials has to be used [49]. The person must transform sensory information into motor responses, ensuring that their movements align with external demands. Individuals can adapt to the context even if the movement is not sufficiently accurate but it meets the external requirements. The ability to imitate transitive movements leads to greater daily functioning [50], thereby contributing to improving the perception of domain-specific self-efficacy for managing symptoms.

Several dimensions of the ULA explained the self-efficacy for managing cognitive symptoms; that is, showing a lower number of apraxic errors in imitation and ADL performance may lead to a better perception of self-efficacy when facing cognitive alterations due to stroke. Moreover, worse performance in the pantomime of gestures seems to be related to an increase in cognitive self-efficacy perception. These controversial findings between dimensions can be explained because imitation and pantomime involve different sensory pathways in the reception of the information from the external context [51]. In imitation, the patient must observe and replicate a movement precisely, while in pantomime, the patient is asked verbally to perform a gesture. When a gesture is asked for verbally, the patient must mentally imagine it. If a post-stroke patient experiences difficulties in imitation, this fact can lead to feelings of frustration or incompetence. Nevertheless, if apraxic errors occur when the individual performs a pantomime, it may indicate that the mental representation of the movement is imprecise. This cognitive process may generate a greater awareness of deficits compared to imitation. If individuals are aware of these difficulties, it can enhance their perception of self-efficacy in controlling cognitive symptoms.

The association of ULA with self-efficacy perception can be translated into various potential applications in both clinical and research areas. From the clinical setting, these applications may include the need for (i) incorporating self-efficacy assessment in individuals with stroke and apraxia; (ii) implementing strategies to improve stroke symptom management abilities; and (iii) guiding healthcare professionals in designing personalized management of self-efficacy perception in this population. On the other hand, from the research area, the practical application may involve (i) a better understanding of the relationship between praxic function and self-efficacy; (ii) the opening of future research in the early identification of the perceptual sphere regarding the patient’s interaction with the environment through movements after a stroke; and (iii) the importance of including ULA as a relevant construct in the design of studies involving patients with low levels of self-efficacy.

Some limitations may be identified in this study: first, the sample was limited to patients from a single Spanish province, which could reduce the generalizability of the findings to a broader population; second, data about the professional work of the study participants were not collected, which could have provided valuable insights into characterizing the sample; third, the study was based on a cross-sectional design, which restricted the ability to observe changes over time and identify longitudinal patterns in the relationship between ULA and self-efficacy.

## 5. Conclusions

The findings of this study identified significant associations between self-efficacy dimensions and the ULA dimensions. Specifically, general self-efficacy and self-efficacy for emotional symptoms were positively associated with apraxic errors in objectless gesture imitation. Self-efficacy for managing global and social interaction symptoms was influenced by the alteration of transitive gesture imitation. Additionally, self-efficacy for cognitive symptoms was related to apraxic error performance in activities of daily living performance, intransitive imitation, and meaningless pantomime. These results highlight the importance of considering the dimensions of apraxia when approaching perceived self-efficacy in post-stroke patients.

## Figures and Tables

**Figure 1 healthcare-11-02252-f001:**
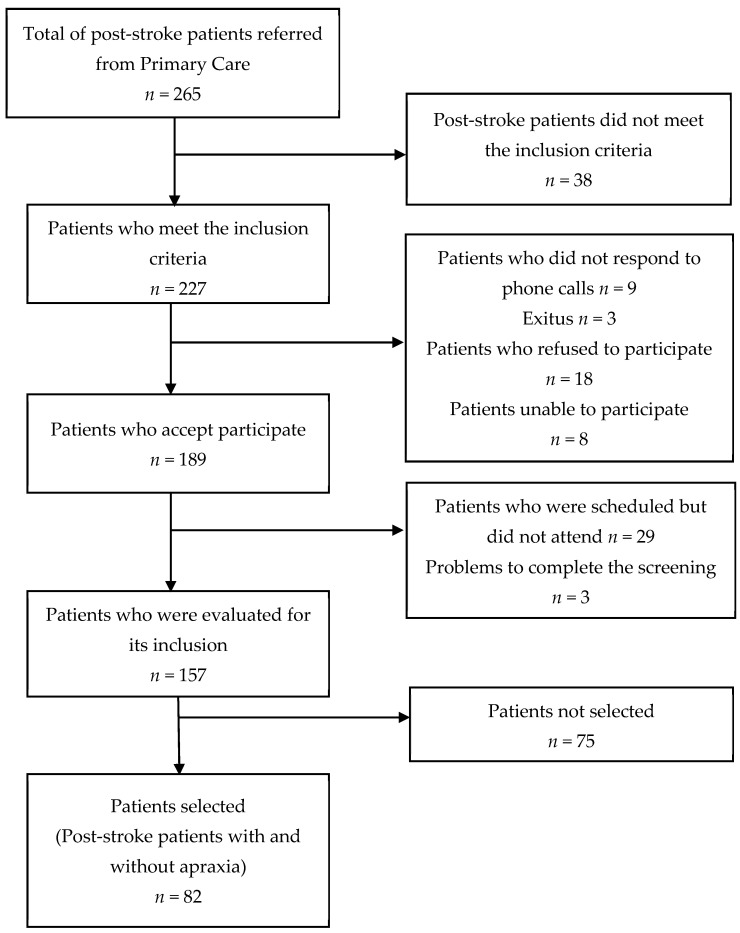
Flowchart of the study population.

**Table 1 healthcare-11-02252-t001:** Comparison of sample descriptive data between participants with and without ULA (n = 82).

Sample Descriptive Data	ULA (*n* = 41)	Non-ULA (*n* = 41)	*p*-Value
**Socio-demographic characteristics**			
Age, mean (years)	64.62 ± 12.30	60.60 ± 10.45	0.381
Gender, male, *n*	32 (78%)	28 (68%)	0.319
Marital status			
Single, *n*	6 (14.6%)	3 (7.3%)	0.678
Married, *n*	30 (73.2%)	31 (75.6%)	
Divorced, *n*	3 (7.3%)	5 (12.2%)	
Widower, *n*	2 (4.9%)	2 (4.9%)	
Occupation			
Employee, *n*	26 (63.4%)	33 (80.5%)	0.177
Retiree, *n*	14 (34.1%)	8 (19.5%)	
Student, *n*	1 (2.4%)	0	
Educational Level			
Primary education, *n*	20 (48.8%)	12 (29.3%)	0.205
Secondary education, *n*	8 (19.5%)	13 (31.7%)	
Vocational training, *n*	4 (9.8%)	8 (19.5%)	
University education, *n*	9 (22%)	8 (19.5%)	
**Self-reported lifestyle behaviours**			
Smoking, *n*	6 (15%)	9 (22%)	0.391
Alcohol consumption, *n*	11 (26.8%)	11 (26.8%)	0.947
Caffeine consumption, *n*	11 (26.8%)	17 (41.5%)	0.286
Hours of sleep, mean	6.71 ± 1.95	6.78 ± 1.40	0.044
Physical activity, mean (minutes per day)	42.41 ± 51.62	61 ± 59.44	0.889
**Clinical characteristics**			
Dominant hand, right, *n*	36 (87.8%)	37 (90.2%)	0.545
BMI, mean (kg/m^2^)	27.50 ± 4.92	28.73 ± 4.98	0.910
OT received after stroke, *n*	4 (9.8%)	6 (14.6%)	0.500
OT rehabilitation, mean (days)	18.56 ± 86.69	35.05 ± 172.39	0.357
Hemiplegia, *n*	2 (4.9%)	2 (4.9%)	1.000
**Stroke characteristics**			
Stroke type			
Ischemic stroke, *n*	37 (90.2%)	34 (82.9%)	0.331
Haemorrhagic stroke, *n*	4 (9.8%)	7 (17.1%)	
Stroke hemisphere			
Right, *n*	17 (41.5%)	14 (34.1%)	0.421
Left, *n*	17 (41.5%)	15 (36.6%)	
Indeterminate, *n*	7 (17.1%)	12 (29.3%)	
Months since stroke, mean	29.54 ± 9.62	32.76 ± 21.12	0.132

Note. ULA = upper limb apraxia; BMI = body mass index; OT = occupational therapy.

**Table 2 healthcare-11-02252-t002:** Descriptive statistics for ULA dimensions (n = 82).

Dimensions of ULA	Mean	SD	Range	Minimum	Maximum	Percentiles		
**TULIA test**						**25th**	**50th**	**75th**
Non-symbolic imitation	35.96	3.14	16	24	40	35.00	37.00	38.00
Intransitive imitation	36.38	3.23	14	26	40	35.00	37.00	39.00
Transitive imitation	29.55	5.55	26	14	40	27.00	30.00	33.00
Non-symbolic pantomime	35.16	3.70	16	24	40	33.00	36.00	38.00
Intransitive pantomime	36.68	3.11	13	27	40	35.00	38.00	39.00
Transitive pantomime	32.74	4.62	20	20	40	30.75	33.00	36.00
**ADL observation scale**								
Total scale score	2.32	4.36	26	0	26	0.00	0.00	3.00

Note. ULA = upper limb apraxia; SD = standard deviation; ADL = activities of daily living.

**Table 3 healthcare-11-02252-t003:** Multiple linear regression models for each self-efficacy dimension (n = 82).

**General Self-Efficacy (R^2^ = 0.138)**						
**Independent Variable**	**B**	**95% CI**		**β**	** *SE* **	***p*-Value**
		**Lower** **Bound**	**Upper** **Bound**			
Intransitive imitation	1.743	0.782	2.704	0.372	0.483	<0.001
**Social–Home Integration Self-Efficacy (R^2^ = 0.052)**						
**Independent Variable**	**B**	**95% CI**		**β**	** *SE* **	***p*-Value**
		**Lower** **Bound**	**Upper** **Bound**			
Transitive imitation	0.263	0.014	0.512	0.229	0.125	0.035
**Cognitive Self-Efficacy (R^2^ = 0.248)**						
**Independent Variable**	**B**	**95% CI**		**β**	** *SE* **	***p*-Value**
		**Lower** **Bound**	**Upper** **Bound**			
Intransitive imitation	1.469	0.660	2.278	0.420	0.406	0.001
ADL total	−0.722	−1.261	−0.183	−0.279	0.271	0.009
Non-symbolic pantomime	−0.783	−1.480	−0.086	−0.256	0.350	0.028
**Emotional Self-Efficacy (R^2^ = 0.104)**						
**Independent Variable**	**B**	**95% CI**		**β**	** *SE* **	***p*-Value**
		**Lower** **Bound**	**Upper** **Bound**			
Intransitive imitation	0.865	0.301	1.429	0.323	0.283	0.003
**Total Score of Self-Efficacy for Managing Symptoms (R^2^ = 0.124)**						
**Independent Variable**	**B**	**95% CI**		**β**	** *SE* **	***p*-Value**
		**Lower** **Bound**	**Upper** **Bound**			
Transitive imitation	1.282	0.524	2.039	0.352	0.381	0.001

Note. R^2^ = regression coefficient of determination; B = regression coefficient; CI = confidence interval; β = adjusted coefficient from multiple linear regression analysis; SE = coefficient standard error; ADL = activities of daily living.

## Data Availability

The data presented in this study are available on request from the corresponding author.

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
