# Peer review of "The Impact of Upper Limb Apraxia on General and Domain-Specific Self-Efficacy in Post-Stroke Patients"

_healthcare, 2023, doi:10.3390/healthcare11162252_

Round 1

Reviewer 1 Report

If the TULIA test is divided into 7 dimensions, not all data are shown in Table 2. Why?

If the objective is to show the impact of upper extremity apraxia in postictus patients, why is a group of patients without apraxia included?

On the other hand, Table 2 shows the Regression Model for one of the groups in the study. Which group does it refer to? What differences are there in the scores for the other group?

Results data are included in the discussion section.

What practical application do the results presented have at the clinical and/or research level?

Regards,

Reviewer 2 Report

The aim of the study was to examine if upper limb apraxia dimensions are related to the levels of  general perceived self-efficacy and self-efficacy for managing symptoms among post- stroke patients. The introduction justifies the research problem correctly. The methodology comprehensively describes the tools used and how the study was conducted. I suggest that the description be supplemented with information on the order of testing with the various tools and information on the researchers carrying out the measurements, their competence, training, knowledge related to participation in the study, etc.  The results were presented correctly and briefly discussed. I would have supplemented them with the results of the TULIA test so that the reader would know more accurately what degree of apraxia the study participants exhibited. In the comparison between groups in Table 1, I suggest supplementing the NIHSS test results as well. The authors also did not show the results of the general self-efficacy scale tests. I wonder if post-stroke subjects without apraxia differed significantly here from those with apraxia?  What I find missing from the discussion is a reference to the professional work undertaken by many of the study participants, how does the identified apraxia translate into the specifics of that work? The discussion also lacks for me a summary of how the results obtained translate into practice, as indicated by the correlations found. What forms of therapeutic interventions could improve the quality of life of people after stroke?  There is also no indication of the limitations of the self-study. 

Round 2

Reviewer 1 Report

Good afternoon,

Thank you very much for the included modifications.

Congratulations on your work!

Greetings,